# Dissolving Microneedles Loaded with Nanoparticle Formulation of Respiratory Syncytial Virus Fusion Protein Virus-like Particles (F-VLPs) Elicits Cellular and Humoral Immune Responses

**DOI:** 10.3390/vaccines11040866

**Published:** 2023-04-18

**Authors:** Ipshita Menon, Smital Patil, Priyal Bagwe, Sharon Vijayanand, Akanksha Kale, Keegan Braz Gomes, Sang Moo Kang, Martin D’Souza

**Affiliations:** 1Center for Drug Delivery Research, Vaccine Nanotechnology Laboratory, College of Pharmacy, Mercer University, Atlanta, GA 30341, USA; 2Center for Inflammation, Immunity & Infection, Institute for Biomedical Sciences, Georgia State University, Atlanta, GA 30303, USA

**Keywords:** vaccine, dissolving microneedles, RSV, fusion protein, nanoparticles, PLGA, virus-like particles, mucosal, humoral, cellular immune response

## Abstract

Respiratory syncytial virus (RSV) is one of the leading causes of bronchiolitis and pneumonia in children ages five years and below. Recent outbreaks of the virus have proven that RSV remains a severe burden on healthcare services. Thus, a vaccine for RSV is a need of the hour. Research on novel vaccine delivery systems for infectious diseases such as RSV can pave the road to more vaccine candidates. Among many novel vaccine delivery systems, a combined system with polymeric nanoparticles loaded in dissolving microneedles holds a lot of potential. In this study, the virus-like particles of the RSV fusion protein (F-VLP) were encapsulated in poly (D, L-lactide-*co*-glycolide) (PLGA) nanoparticles (NPs). These NPs were then loaded into dissolving microneedles (MNs) composed of hyaluronic acid and trehalose. To test the in vivo immunogenicity of the nanoparticle-loaded microneedles, Swiss Webster mice were immunized with the F-VLP NPs, both with and without adjuvant monophosphoryl lipid A (MPL) NPs loaded in the MN. The mice immunized with the F-VLP NP + MPL NP MN showed high immunoglobulin (IgG and IgG2a) levels both in the serum and lung homogenates. A subsequent analysis of lung homogenates post-RSV challenge revealed high IgA, indicating the generation of a mucosal immune response upon intradermal immunization. A flowcytometry analysis showed high CD8+ and CD4+ expression in the lymph nodes and spleens of the F-VLP NP + MPL NP MN-immunized mice. Thus, our vaccine elicited a robust humoral and cellular immune response in vivo. Therefore, PLGA nanoparticles loaded in dissolving microneedles could be a suitable novel delivery system for RSV vaccines.

## 1. Introduction

Respiratory syncytial virus (RSV) is one of the main causes of pulmonary disease in infants, toddlers, and the elderly (above the age of 65) [1]. The lack of a marketed vaccine during the latest RSV outbreak has highlighted the strain that RSV can have on healthcare infrastructure [2,3]. Early vaccine candidates for RSV included inactivated virus adsorbed on alum. This vaccine, however, did not protect the vaccinated toddlers against the infection, instead leading to exacerbated respiratory disease [4]. It was later found that the formalin inactivation damages the fusion (F) protein, which is the main antigenic epitope on the RSV [5]. The F-protein is a highly conserved protein among both strains of RSV, such as RSV A and B. To make the F-protein more immunogenic, it can be genetically engineered to form a virus-like particle (VLP) that mimics the structure of the virus while remaining a nonvirulent F-VLP antigen [6].

In this study, we encapsulated the F-VLP antigen in a polymeric matrix to improve its immunogenicity [7]. The polymeric nanoparticle matrix also helps prevent the degradation of the vaccine antigen in the body. Moreover, the size of the polymetric nanoparticles can be formulated to resemble the size of pathogens to improve the uptake by antigen-presenting cells [8]. Particulate vaccines have been proven to induce cytotoxic T lymphocyte responses, which may be beneficial in viral infections [9]. Furthermore, polymeric nanoparticles consisting of sustained-release polymers can release the antigen gradually over time by slow erosion of the matrix over an extended period. The main advantage of a slow release of the payload is the reduced need for multiple booster doses of the vaccine. This sustained release of antigens has also been proven to enhance memory responses such as T-cell memory [10]. One of the reasons that led to the failure of the RSV vaccine study in the 1960s was that the formalin-inactivated virus resulted in a more Th2-skewed immune response and lacked the induction of a cytotoxic T-cell response [5]. A nanoparticle delivery system also allows for the incorporation of immunomodulatory adjuvants to boost and tailor the type of immune response [8]. Adjuvant nanoparticles and antigen nanoparticles in separate nanoparticles can also be used to great effect [11]. Thus, in this study we utilized monophosphoryl lipid A, which is a TLR4 agonist, as the adjuvant. Such an adjuvant will help skew the immune response towards the Th1 arm [12,13].

In addition to the vaccine antigen, the route of administration of the vaccine has a great impact on the immune response generated and the patient compliance [14]. A large number of individuals associate vaccination with the pain that it comes along with [14,15]. This results in vaccine hesitancy. Moreover, most of the vaccines under trial for RSV also utilize the intramuscular route of administration [16,17]. The intradermal route would be a less painful alternative to the hypodermic needle. Additionally, the skin provides an abundant presence of Langerhans cells, which are specialized antigen-presenting cells (APCs) that can take up antigens and present them to the immune organs [15,18,19,20]. An attractive method for skin administration involves the use of microneedles. Microneedles are tiny micron-sized needles arranged on a patch. There are several types of microneedles, such as solid microneedles and hollow microneedles; however, dissolving microneedles is one of the most sought-after methods of intradermal vaccination due to the fact of their versatility in loading different kinds of antigens [21]. Moreover, dissolving microneedles can also be loaded with different nanoparticles that will simply be released in the skin upon application [18,22,23,24]. We have previously shown that when the F-VLP antigen is encapsulated into a PLGA polymeric nanoparticle matrix, it improves the immunogenicity in vitro [25]. Therefore, in this proof-of-concept study, we demonstrate how loading F-VLP nanoparticles along with adjuvant MPL-A nanoparticles into dissolving microneedles induces a robust humoral and cellular immune response in vivo.

## 2. Materials and Methods

Poly(lactic-co-glycolic) acid (PLGA) (70:25) was purchased from Evonik Industries (Essen, Germany). Dichloromethane (DCM) was purchased from Fisher Scientific. Albumin–fluorescein isothiocyanate conjugate (FITC BSA), trehalose dihydrate, sorbitan monooleate/Span^®^ 80, methylene blue, and hematoxylin and eosin (H&E) reagents were purchased from Sigma Aldrich (St Louis, MO, USA). Sodium hyaluronate (HA) (5 kD, 50 kD, 150 kD, and 500 kD) was purchased from Lifecore Biomedical (Chaska, MN, USA). The 8 × 8 array silicone microneedle templates were obtained from Micropoint Technologies (Singapore). The Pierce Micro BCA Assay Kit was purchased from Thermo Fisher (Waltham, MA, USA). The RSV A2 virus and wild-type RSV F-VLP were previously well described in detail in a prior study [26]. Briefly, Spodoptera frugiperda (Sf9) cells were used to express RSV-F VLP vaccines after infection with recombinant baculovirus-expressing RSV-F and M1, as previously reported [26]. The following reagent was obtained through BEI Resources, NIAID, NIH: F-Protein with C-Terminal Histidine Tag from Respiratory Syncytial Virus, B1, Recombinant from Baculovirus, NR-31097. Six- to eight-week-old Swiss Webster female mice were purchased from Charles River Laboratories (Wilmington, MA, USA). HRP-tagged secondary goat anti-mouse IgG, IgA, IgG2a, and IgG1 were purchased from Invitrogen (Waltham, MA, USA). Anti-mouse antibodies, such as fluorescein isothiocyanate (FITC)-labeled CD8 (clone: 53-6.7) and allophycocyanin-labeled CD4 (clone: GK 1.5), which were used for flow cytometry analysis, were purchased from eBioscience laboratories (San Diego, CA, USA) and BioLegend (San Diego, CA, USA).

### 2.1. Formulation of F-VLP-Loaded Polymeric Nanoparticles

The formulation, characterization, and in vitro cytotoxicity evaluation and in vitro immunogenicity testing of the F-VLP PLGA nanoparticles were carried out as we published earlier [25]. Briefly, PLGA nanoparticles were formulated by preparing a W/O/W double emulsion. The resulting double emulsion was passed through a high-pressure homogenizer to achieve the size reduction of the nanoparticles. Next, the solvent was removed by constant stirring for 5 h. Finally, the nanoparticles were washed with phosphate buffer and then resuspended in 2% trehalose dihydrate and freeze dried to obtain the dried nanoparticles.

### 2.2. Formulation of Dissolving Microneedles

The microneedles were prepared using polydimethylsiloxane (PDMS) molds. The dissolving microneedles (MNs) were composed of hyaluronic acid (HA) and trehalose dihydrate (trehalose). Different grades of HA were screened to create the MNs (5 kD, 50 kD, 150 kD, and 500 kD). The grade that formed strong microneedles was chosen to make the dissolving MNs. Once, the grade of HA was chosen (150 kD), blank dissolving MNs using different concentrations of HA and trehalose were prepared (Table 1). Briefly, a gel of 10% *w*/*v* of HA and 5% *w*/*v* of trehalose was made using deionized water. Ten milligrams of the gel were added onto the PDMS mold to cover the entire surface and centrifuged at 4000 rpm for 15 min at 15 °C in a swinging basket centrifuge (Thermo Scientific, Waltham, MA, USA). The mold was then air-dried in a desiccator for 5–6 h till partially dried. A backing layer comprising 20% *w*/*v* of HA was added, and the mold was completely air-dried in a desiccator overnight. The next day, the MN patch was demolded using a 3M double-sided tape (3M, Saint Paul, MN, USA). The nanoparticle-loaded MNs were prepared by suspending the nanoparticles in the gel before adding onto the mold. This step was repeated 2–3 times to ensure that the microneedle tips were filled with the nanoparticles.

### 2.3. Characterization of the Nanoparticle-Loaded Dissolving Microneedles

The F-VLP NP-loaded dissolving MNs were imaged using scanning electron microscopy (SEM) before and after application on murine skin. The microneedle patch was stuck on metal stubs with double-sided adhesive carbon conductive tabs. The images were captured at different magnifications using a Phenome benchtop SEM (Nanoscience Instruments, Phoenix, AZ, USA) (Bansal et al., 2020). The strength of the microneedles was tested using a stack of 5 laboratory parafilms (Parafilm“M”Laboratory film, Neenah, WI, USA). The microneedle patch was deemed strong if it pierced all five layers of the parafilms (Nguyen et al., 2018). The microchannels created by the microneedles upon application to murine skin was visualized by imaging using a digital microscope (Plugable© Digital Viewer) after staining with 1% *w*/*v* of methylene blue. A histology of the skin was performed after embedding in TissueTek^®^, O.C.T. Compound (Sakura Finetek USA, Torrance, CA, USA). Sections of 10 μm thickness were obtained using a cryostat microtome (Microm HM 505 E Cryostat, Ramsey, MN, USA) and stained with hematoxylin and eosin (H&E) to observe the histological changes on the skin upon treatment with the dissolving microneedles [27]. The images of the stained sections were captured using a Leica DM 750 microscope (Leica Microsystems, Wetzlar, Germany). To quantify the antigen content in the microneedle patch, the microneedle tips were cut off using a scalpel. The needle tips and the microneedle base were then separately dissolved in microcentrifugation tubes containing 1 mL of PBS. The tubes were centrifuged to pellet the nanoparticles loaded in the microneedles. The content of antigen was analyzed using a micro-BCA protein assay, as per manufacturer’s instructions.

### 2.4. Immunization with Nanoparticle-Loaded Dissolving MNs and RSV Challenge

Six-eight-week-old Swiss Webster mice were divided into five study groups (*n* = 5): naïve (untreated) (negative control); F-VLP suspension group—20 μg F-VLP suspension (20 μg of F-VLP-loaded dissolving MN); blank NP MN group (blank NP-loaded dissolving MN); F-VLP NP MN group—20 μg F-VLP (F-VLP NP-loaded dissolving MN); and F-VLP NP + MPL NP group—20 μg F-VLP and 5 μg MPL (F-VLP NP + MPL NP-loaded dissolving MN). Before immunization the hair on the dorsal side of the mice was removed using Nair depilatory cream (Newing, NJ, USA). The mice received one prime and one booster dose at an interval of three weeks. Blood was collected at weeks two, four, and six to analyze the F-protein-specific serum antibody levels. Seven weeks after the final boost, the mice were challenged with RSV A2 (1 × 10^6^ PFU) administered intranasally: 25 µL per nostril under isoflurane anesthesia (Isothesia, Melville, NY, USA). One week later, the mice were sacrificed by carbon dioxide inhalation, and their spleen, lymph nodes, and lungs were harvested for further analysis. All studies were carried out in accordance with Mercer University’s IACUC (A1504008).

### 2.5. Measurement of Antigen-Specific Antibodies in Serum and Lung Homogenates Using ELISA

The RSV-specific IgG antibodies and the antibody subtypes (IgG1 and IgG2a) were analyzed using an enzyme-linked immunosorbent assay (ELISA) using F-protein (BEI Resources) as the coating antigen [26,28,29,30]. Briefly, 96-well microplates (Nunc, Rochester, NY, USA) were coated with F-protein (200 ng/well) and incubated overnight at 4 °C. The plates were then washed with PBST (PBS with 0.05% Tween 20) and followed by a blocking step with 3% BSA for 90 min at 37 °C. The serum or lung homogenates were serially diluted with PBS and added to the wells followed by incubation for 90 min at 37 °C. Next, the wells were washed thrice with PBST, after which the secondary antibody conjugated to horseradish peroxidase (HRP) IgG, IgG1, IgG2a (serum and lung), and IgA (lung) (Invitrogen, Waltham, MA, USA), at a 1:2000–4000-fold dilution, was added to the wells. After incubation for 90 min at 37 °C, the plate was washed three times with PBST followed by the addition of the substrate tetramethybenzidine (TMB) (BD Bioscience, San Jose, CA, USA). The color reaction was stopped using 0.3 M sulfuric acid after incubating the TMB for 15 min. The optical density (OD) at 450 nm was measured using a Synergy HI microplate reader (BioTek, Winooski, VT, USA).

### 2.6. Measurement of T Cells in the Immune Organs Using Flow Cytometry

After RSV challenge, the mice were sacrificed, and their brachial and inguinal lymph nodes and spleens were harvested. The lymph nodes and spleen were crushed and passed through a 40 µm cell strainer to prepare the single cell suspensions. To remove the red blood cells from the spleen, the cells were treated with ammonium chloride potassium (ACK) lysis buffer for three minutes. After processing, the cells were stored at −80 °C until they were analyzed. Before analyzing the cells for T-cell markers, the cell count of the viable cells was analyzed by the trypan blue exclusion method using a Countess II FL Automated Cell Counter (Invitrogen, Waltham, MA, USA). After diluting the cell suspensions to 1 × 10^6^ cells, the lymph node and spleen cells were stained with fluorescent-tagged antibodies specific for T-cell surface markers CD4 and CD8 (eBioscience laboratories, San Diego, CA, USA). The cells were incubated on ice for one hour and then washed with PBS to remove the excess unbound marker. The samples were then suspended in PBS and for each sample 10,000 events/cells were acquired using a BD Accuri C6+ flow cytometer (BD Bioscience, San Jose, CA, USA).

### 2.7. Statistical Analysis

All experiments were performed in triplicate unless otherwise indicated. The data analyses were performed using GraphPad Prism version 9.5.1 and expressed as the standard error of mean (SEM). The normal distribution of the data was analyzed using the Shapiro–Wilk test. The one-way ANOVA and post hoc Tukey tests were used for multiple comparisons between groups. The two-way ANOVA and post hoc Tukey tests were used among between groups and between weeks (* *p* < 0.05, ** *p* < 0.01, and *** *p* < 0.001).

## 3. Results

### 3.1. Characterization of Nanoparticle-Loaded Microneedles

Seven different microneedle formulations were prepared using different concentrations of HA (150 kDA) and trehalose. The microneedles made with just HA took more time to dissolve. The addition of trehalose helped in improving the time taken by the needles to dissolve; however, a higher concentration of trehalose also caused the microneedles to become sticky. The observations made upon the preparation of the different formulations of microneedles are described in Table 1. The microneedles from formulation #4 were 404 µm in length, as observed in in the SEM image shown in Figure 1, and were strong enough to penetrate up to four layers of Parafilm^®^ M, as represented in Figure 2. The microneedles dissolved within 10 min and created aqueous micropores, which were stained and visualized by methylene blue, which is an aqueous dye. The H&E-stained transverse section of the skin shown in Figure 3 depicts a micropore that had penetrated the stratum corneum and the epidermis. The micro-BCA protein assay showed that the antigen-loading efficiency in the microneedle tips was 54% ± 5.5.

### 3.2. RSV F-Protein-Specific IgG and Subtype Antibodies Observed in the Serum of Vaccinated Mice

We observed a significant amount of F-protein-specific IgG in the serum of mice that were immunized with the F-VLP NP + MPL NP MN (Figure 4) compared to the mice vaccinated with the F-VLP suspension MN. The serum IgG was observed two weeks after the prime dose. The mice treated with the F-VLP NP MN did not show significantly high antibody levels compared to the F-VLP suspension MN group. The high antibody levels in the F-VLP NP + MPL NP MN group was maintained until week 6 compared to the other groups (Figure 4). An analysis of the IgG subtypes using ELISA showed that the IgG1 levels were significantly high for the F-VLP NP MN + MPL NP MN group compared to the naïve mice at week 6 compared to week 4. The F-VLP NP MN + MPL NP MN, F-VLP NP MN, and F-VLP suspension MN groups did not show significant levels of IgG1. However, we observed significantly high levels of IgG2a in the F-VLP NP MN group and F-VLP NP + MPL NP MN group when compared to the F-VLP suspension MN group in both weeks 4 and 6.

### 3.3. RSV F-Protein-Specific IgG, IgG Subtypes, and IgA Antibodies Observed in the Lung Homogenates of Vaccinated Mice after RSV Challenge

We observed significantly high levels of IgG and IgA in the lung homogenates of mice immunized with the F-VLP NP + MPL NP MN compared to the F-VLP suspension MN group. The F-VLP NP MN group had higher IgG compared to the F-VLP suspension MN group. We observed significantly high levels of IgG2a in the lung homogenates of mice immunized with the F-VLP NP + MPL NP MN compared to the F-VLP suspension and F-VLP NP MN groups (Figure 5). The IgG1 subtype levels in the lung homogenates were not significant in any of the treatment groups.

### 3.4. Enhanced Expression of T-Cell Markers Observed in Spleen and Lymph Nodes of Vaccinated Mice

Heightened induction of CD8+ T cells was observed in both the lymph node and spleen cells of the mice vaccinated with the F-VLP NP + MPL NP MN compared to the unvaccinated naïve mice and the mice vaccinated with the F-VLP suspension MN (Figure 6). The induction of CD4+ cells was significantly higher in the lymph node cells of the mice vaccinated with the F-VLP NP + MPL NP MN when compared to the naïve mice and the F-VLP suspension MN group. In spleen cells, the expression of CD4+ cells was high in the F-VLP suspension MN group compared to the naïve mice. The CD4+ cells in the lymph node cells of the F-VLP NP + MPL NP MN group and the F-VLP NP MN group were nonsignificant when compared to the F-VLP suspension MN group. However, the lymph node cells of the F-VLP NP + MPL NP MN group and the F-VLP NP MN group showed a higher CD4+ T-cell population when compared to the blank NP MN and naïve group.

## 4. Discussion

We chose dissolving microneedles as the delivery platform to study the in vivo efficacy of our vaccine. We optimized the formulation of the dissolving microneedles by varying different concentrations of HA and trehalose. We observed that the microneedles made with low-molecular-weight (5 kDa) and very high-molecular-weight (500 kDA) HA did not form microneedles. The HA with a molecular weight of 5 kDA and 50 kDA formed microneedles; however, when they were demolded from the template, the microneedle patch broke when we tried to demold it as one single patch. Thus, the 5 kDA and 50 kDA HA formed microneedles that were very brittle. The HA with a molecular weight of 500 kDA formed a gel that was very viscous; thus, it was difficult to cast the gel on the template. The thickness of the gel also impeded the formation of microneedles. This may be due to the inability of the gel to fill the microneedle template. Thus, the 5 kDa, 50 kDA, and 500 kDA were not chosen to prepare the microneedles. Finally, we observed that microneedles formed with 150 kDA molecular weight HA formed the best microneedles, which were strong enough to penetrate four layers of Parafilm^®^ M. The addition of trehalose reduced the time needed to dissolve the microneedles. Different concentrations of trehalose and HA were tested, and as the concentration of HA increased, the time needed to dissolve increased; as the concentration of trehalose increased, the time needed to dissolve the needles decreased. However, significantly increasing the concentration of trehalose also caused the microneedles to become sticky and did not allow for the formation of sharp needles. The final formulation consisted of 10% *w*/*v* of HA and 5% *w*/*v* of trehalose. These microneedles dissolved in 10 min, formed micropores on the surface of the mice skin, and were able to breach the stratum corneum. These microneedles did not penetrate five layers but penetrated four layers of Parafilm^®^ M, which was approximately 0.52 mm deep, as each Parafilm^®^ M is 0.13 mm in thickness [31]. The dermis of the mouse skin is 0.62 mm and that of human skin is approximately 0.512 to 1.977 mm, as per the literature [32]. Thus, this needle penetration depth will be suitable for both mouse and human skin for vaccination.

The fusion (F) glycoprotein is the protein that is essential for the fusion of the virus with the host cell, thus making the F-protein a suitable vaccine candidate. In this study, virus-like particles of full-length F-protein were used as the antigen [30]. The F-VLP was encapsulated in a polymeric matrix made of PLGA, which leads to the sustained release of the antigen (published previously) [25]. Such a sustained release of antigen will help release the antigen over a long period of time and will, in turn, assist in the generation of a CD8+ cytotoxic T-cell response, which will be beneficial against RSV [33]. The PLGA matrix used is biodegradable and a synthetic polymer; thus, it will not meddle with the immune response generated [8]. The PLGA nanoparticles of the F-VLP antigen improves the antigen presentation of the F-VLP to antigen-presenting cells such as dendritic cells (published previously) [25]. The PLGA matrix also allows for the incorporation of adjuvants [9,34]. We hypothesized that the improved antigen presentation will lead to the enhanced presentation of the F-VLP antigen to the T cells and B cells [7,35]. This, subsequently, will lead to an enhanced humoral and cellular immune response in a preclinic mouse model for RSV. We evaluated this hypothesis in this proof-of-concept in vivo study. For the in vivo study, before the application of the nanoparticle-loaded microneedle vaccine, we shaved the fur from the dorsal region of the mouse to avoid the mice fur from interfering with the microneedle application. The microneedles were applied a day after the hair removal to provide the mice skin some time to recover from the hair removal process. The hair removal, using depilatory cream, caused minimal redness which improved in a few hours. However, this would not affect the immune response regenerated because the immune cells recruited due to the minimal inflammation will not be specific to the vaccine antigen.

The optimized dissolving microneedle formulation was used in the delivery of the polymeric nanoparticle-based vaccine via the intradermal route. Delivering the polymeric nanoparticles to the epidermal and dermal layers of the skin will allow for the dermal dendritic cells and Langerhans cells to take up the nanoparticles [36,37,38]. These antigen-presenting cells will then subsequently present them to the T cells in the lymph nodes and spleen. The adjuvant monophosphoryl lipid A (MPL) was utilized to enhance the immune response, as it is known to induce a Th1-polarized immune response, which would be very desirable for RSV [13]. We analyzed the F-protein-specific antibodies in their serum and found increased levels of IgG and the subtype IgG2a in the mice immunized with the F-VLP NP + MPL NP MN vaccine. A similar antibody response was observed in the lung homogenates, indicating the generation of a robust humoral immune response, which may be attributed to the synergistic effect of using polymeric nanoparticles and an intradermal route of administration [7,39]. We observed elevated IgA levels in the lung homogenates of the F-VLP NP + MPL NP MN group, indicating the generation of a mucosal immune response. This may be attributed to the intradermal immunization and the use of adjuvant MPL leading to mucosal homing of the antigen in the lungs [40,41,42,43]. The humoral immune response was corroborated by the T-cell response. The IgG2a helps the stimulation of the CD8+ cytotoxic T cells. We observed the enhanced expression of the CD8+ T cells in both the lymph nodes and the spleen in the F-VLP NP + MPL NP MN group compared to the F-VLP suspension MN group. We also observed the enhanced expression of CD4+ T cells in the lymph node of the F-VLP NP + MPL NP MN group which again supports the antibody results. Thus, the F-VLP NP + MPL NP MN group also elicited a cellular immune response, which is vital for viral infections such as RSV.

The shortcoming of this study is that we studied the immune response only for six weeks and did not look at the presence of memory T and B cells. Future studies will involve looking at the cytokine expression, such as IL4 and IFN-γ, and the presence of memory cells. Additionally, we will study the efficacy of the vaccine by carrying out a serum neutralization assay and evaluating the lung viral titer to better understand the immune response generated by the F-VLP NP + MPL NP MN vaccine.

## 5. Conclusions

In this proof-of-concept study, we were able to show that PLGA NPs loaded in dissolving microneedles can be an efficient platform for the delivery of the F-VLP antigen. We were able to formulate an optimized dissolving microneedle formulation made of hyaluronic acid and trehalose that dissolved completely within ten minutes. The nanoparticle delivery system also allowed to incorporate the adjuvant MPL-A. We observed enhanced antibody levels of IgG and its subtype IgG2a in the serum and lung homogenates after immunization with the F-VLP NP + MPL NP MN vaccine. We also observed IgA, indicating the generation of a mucosal response after immunization with the F-VLP NP + MPL NP MN vaccine. Dissolving microneedles is a convenient, easy-to-use, and painless delivery system. Such a painless system would be valuable as a vaccine delivery platform for children. We also observed the generation of a robust T-cell response due to the presence of CD8+ T cells in the lymph nodes and spleens of the mice vaccinated with the F-VLP NP + MPL NP MN. Thus, in this study, we observed that combining the potential of a nanoparticle-based vaccine with the novel dissolving microneedles resulted in the generation of a robust humoral and cellular immune response. Thus, PLGA NPs of F-VLP loaded in dissolving microneedles can be a potential vaccine candidate for RSV.

## Figures and Tables

**Figure 1 vaccines-11-00866-f001:**
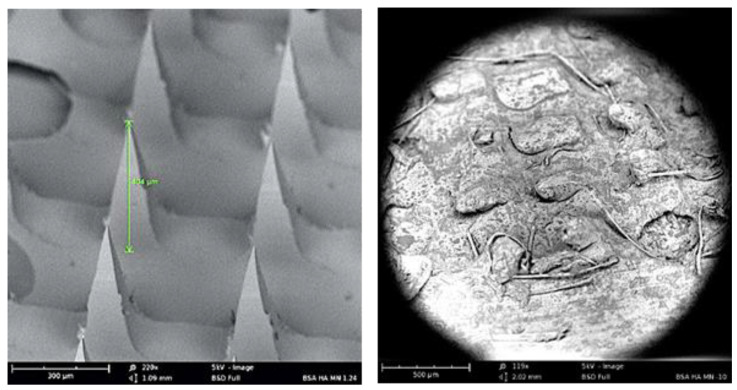
SEM image of dissolving microneedle: (**left**) before application and (**right**) 5 min after application to murine skin.

**Figure 2 vaccines-11-00866-f002:**
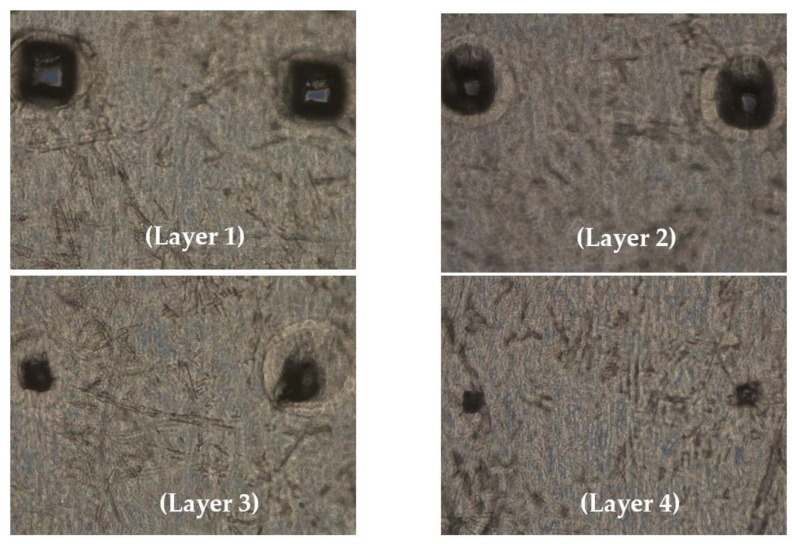
Optical microscopic image of Parafilm M treated with microneedles; the dissolving microneedles penetrated 4 layers of Parafilm^®^ M.

**Figure 3 vaccines-11-00866-f003:**
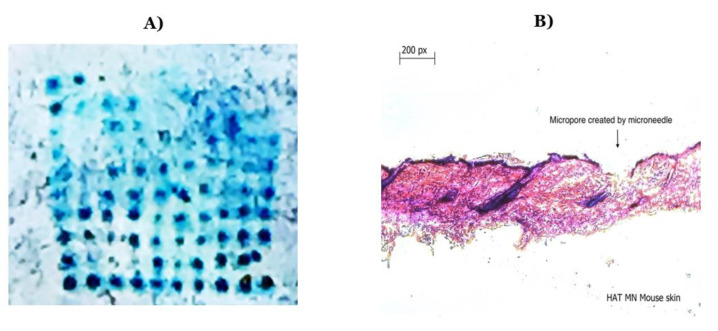
(**A**) Methylene blue staining after the application of microneedles on murine skin; (**B**) hematoxylin and eosin staining of a transverse section of a microneedle applied to murine skin showing a formed micropore.

**Figure 4 vaccines-11-00866-f004:**
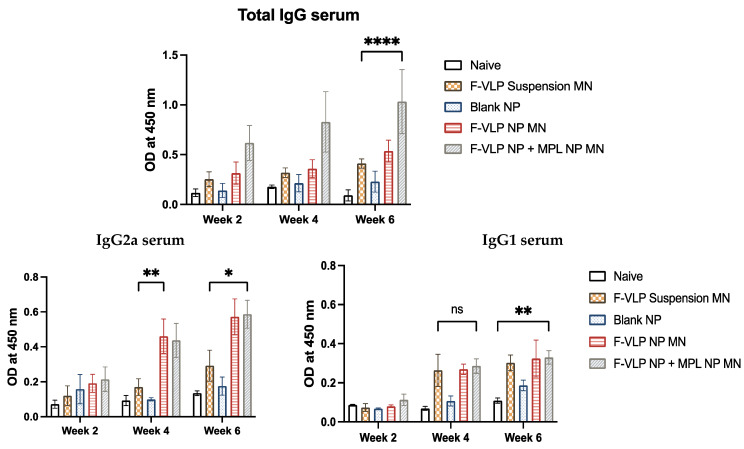
F-protein-specific antibody response in the serum of mice during and after immunization with nanoparticle-loaded microneedles. Serum antibodies specific to the RSV F-protein were analyzed using the RSV F-protein as the ELISA-coating antigen. (Top) F-protein-specific IgG in the serum. (Bottom) F-protein-specific IgG subtypes, IgG2a, and IgG1 in the serum. * *p* < 0.05, ** *p* < 0.01,*** *p* < 0.001 and **** *p* < 0.0001, ns: not significant.

**Figure 5 vaccines-11-00866-f005:**
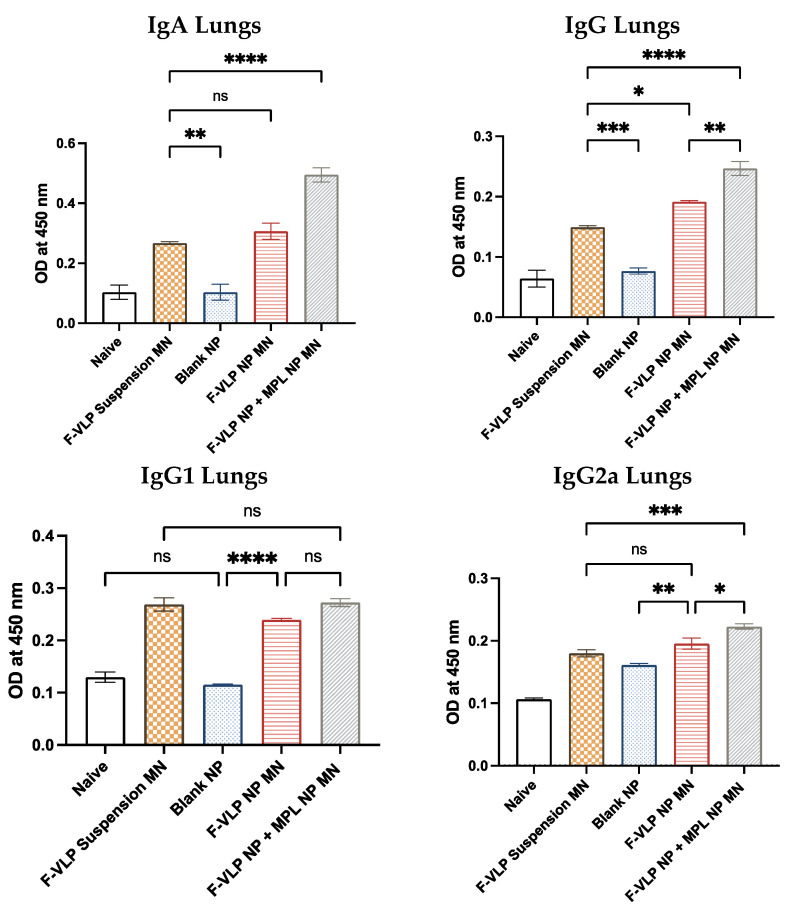
F-protein-specific antibody response in the lung homogenates of immunized mice post-RSV challenge. Antibodies specific to the RSV F-protein were analyzed using the RSV F-protein as the ELISA-coating antigen. (**Top**) F-protein-specific IgG and IgA in the lung homogenates. (**Bottom**) F-protein-specific IgG subtypes IgG2a and IgG1 in the lung homogenates. * *p* < 0.05, ** *p* < 0.01,*** *p* < 0.001 and **** *p* < 0.0001, ns: not significant.

**Figure 6 vaccines-11-00866-f006:**
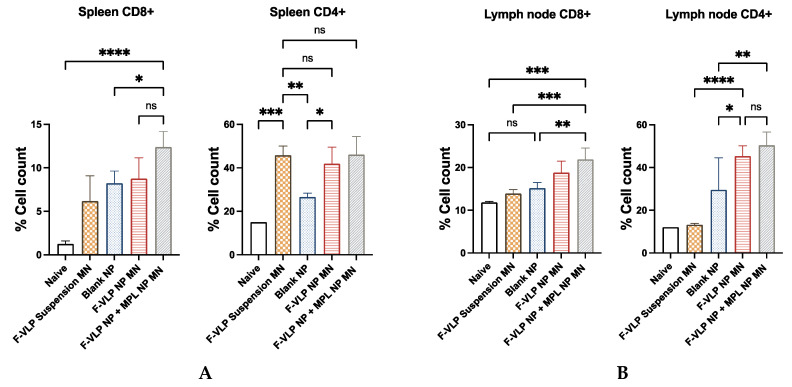
Immunization with intradermal adjuvanted vaccine nanoparticles generated CD8+ and CD4+ T cells: (**A**) CD8+ and CD4+ cells in lymph nodes; (**B**) CD8+ and CD4+ cells in spleen; 10,000 events/cells were acquired for each experiment. * *p* < 0.05, ** *p* < 0.01,*** *p* < 0.001 and **** *p* < 0.0001, ns: not significant.

**Table 1 vaccines-11-00866-t001:** Formulation optimization of the dissolving MNs.

Formulation Number	Hyaluronic Acid (% *w*/*v*)	Trehalose (% *w*/*v*)	Comments	Time Needed to Dissolve Needles (min)
#1	5	10	Needles brittle, soft, pliable base, difficult to demold	15
#2	5	20	Needles brittle, pliable base, difficult to demold	15
#3	5	30	Needles brittle, soft, pliable base, difficult to demold	15
#4	10	5	Needles formed, base nonpliable, easy to demold	10
#5	10	10	Needles formed, base nonpliable, difficult to demold	20
#6	15	5	Needles formed, base nonpliable, easy to demold	15
#7	15	10	Needles formed, base nonpliable, easy to demold	20

## Data Availability

The data presented in this study are available upon request from the corresponding author.

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
