# Peer review of "Dissolving Microneedles Loaded with Nanoparticle Formulation of Respiratory Syncytial Virus Fusion Protein Virus-like Particles (F-VLPs) Elicits Cellular and Humoral Immune Responses"

_vaccines, 2023, doi:10.3390/vaccines11040866_

Round 1

Reviewer 1 Report

The authors of the work "Dissolving microneedles loaded with nanoparticle formulation of respiratory syncytial virus fusion protein virus-like particles (F-VLP) elicits cellular and humoral immune responses" present positive results when demonstrating that their formulation elicits a specific immune response against the RSV virus in mice.

I want to congratulate the authors because the work is very well organized and written. In addition, their results are of great importance for the scenario of respiratory diseases caused by RSV infection. I have just one question and one suggestion.

Question: the animals underwent hair removal from their dorsal region. This procedure may induce skin irritation (inflammatory process). Did the authors verify such a possibility? How much could this inflammatory process have contributed to the success of the immune response induced by the formulation? It would be interesting if the authors briefly reported on this reflection in the text of the article.

Suggestion: in Figure 4, the authors identified the results for IgG2a as "serum IgG2a". Authors must standardize such identification for the IgG and IgG1 results presented in the same figure, that is, add the term "serum" in the identification of the other results.

Author Response

Dear Reviewer,

Thank you for considering our manuscript “Dissolving microneedles loaded with nanoparticle formulation of respiratory syncytial virus fusion protein virus-like particles (F-VLP) elicits cellular and humoral immune responses”, for a revised draft.

We appreciate the time and effort you have devoted to review the manuscript and provide insightful feedback to improve our paper. We have carefully gone through each of the suggestions the reviewers have kindly provided. We hope that edits that we have incorporated in the manuscript address the concerns raised by you.

We have gone through all the comments carefully and worked hard to incorporate the changes and respond to the feedback. We hope our revisions convince you to accept our manuscript.

Please address all correspondence concerning this manuscript to: [email protected]

Sincerely,

Martin D’Souza, Ph.D.

Reviewer 2 Report

The manuscript developed a nanovaccine of RSV F-VLP loaded in dissolving microneedles, and evaluated the level of humoral and cellular immune responses in mouse model. However, many problems remain in this manuscript. Hopefully you can refer to the suggestions below for modifications: 

1.     The formatting problems of the manuscript are relatively serious, which requires careful revision. There are also many inappropriate places or even errors to the picture additionally. Figure 1. The length of 404 μm is too small; Table 1. Duplication; Figure 4. Lower resolution; Figure 5. Presence of irrelevant numbers. Besides that, the format in the figure should be uniform, such as size, order number, etc.

2.     The protective efficacy of the vaccine is not described in this manuscript, please add tissue viral titers and pathological inflammation of the vaccinated mice.

3.     In the process of preparing dissolving microneedles (MN), this manuscript screened different grades of HA to obtain stronger microneedle. Please add the specific data on the relationship between the grades of HA and the effect of microneedles.

4.     This manuscript used 7 different formulations to prepare dissolving microneedles (MN), please add the difference in effect between them instead of just reflecting the results of formulation 4.

5.     During the evaluation of the humoral response to vaccines, this manuscript employed optical density (OD) readings in response to levels of F specific antibodies (IgG/G1/G2a/A) in serum and tissue (Figure 4, 5), which I feel is not very appropriate, please use the dilution factor (ED50).

6.     While assessing the process of competence of a cell's immune response, I wanted to know the levels of cytokines (such as IFN-γ), rather than only displaying the number of CD4+T and CD8+T cells.

7.     Please check carefully on the source of the F protein (NR-31097), as this product is not seen on Bei resources.

Author Response

(The authors gave the same response as above.)

Round 2

Reviewer 2 Report

The references became garbled code, please authors supply plain text type of author response in the future.

Author Response

Clarification regarding the reviewer's comment was requested. It was confirmed by the Editor that the clarification was not provided. Hence, we are submitting the manuscript without addressing the comment.

Thank you